# Constraints on Martian Chronology from Meteorites

**Zoltán Váci** * and **Carl Agee**

Institute of Meteoritics, University of New Mexico, Albuquerque, NM 87131, USA; agee@unm.edu

* Correspondence: zmoney@unm.edu

**Abstract:** Martian meteorites provide the only direct constraints on the timing of Martian accretion, core formation, magmatic differentiation, and ongoing volcanism. While many radiogenic isotope chronometers have been applied to a wide variety of Martian samples, few, if any, techniques are immune to secondary effects from alteration and terrestrial weathering. This short review focuses on the most robust geochronometers that have been used to date Martian meteorites and geochemically model the differentiation of the planet, including $^{147}Sm/^{143}Nd$, $^{146}Sm/^{142}Nd$, $^{176}Lu/^{176}Hf$, $^{182}Hf/^{182}W$, and U-Th-Pb systematics.

**Keywords:** Mars; geochemistry; radiogenic isotopes; geochronology

## 1. Introduction

It is generally agreed that the inner solar system planets were formed by the collisional agglomeration of dust particles into cm- or m-sized objects which then grew into planetesimals up to 100 km in size via turbulent accretion triggered by gravitational instability. Gravitational interactions between planetesimals caused them to collide and produce Mars-sized planetary embryos, which then formed terrestrial planets via giant impacts [1,2]. Under this paradigm, Mars is a leftover planetary embryo which escaped the giant impacts that formed the other terrestrial planets [3]. The timescales of the accretion, core formation, and silicate differentiation of Mars are best constrained using the radiometric isotope systematics of the only available samples from the planet, Martian meteorites.

Since these meteorites are generally hot or cold desert finds that have spent a considerable amount of time exposed to terrestrial surface conditions, they are often heavily weathered [4]. Isotopic chronometers can be highly sensitive to disruption by weathering, and for this reason this short review only focuses on the most pristine and uncompromised chronological data. Emphasis is placed on recent measurements employing the most robust chronometers. Isotopic systems employed include Sm-Nd, Lu-Hf, Hf-W, Ar-Ar, U-Th-Pb, and various noble gas exposure age chronometers such as cosmogenic $^3He$, $^{21}Ne$, and $^{38}Ar$. All ages discussed below are reported with their two sigma uncertainties.

## 2. Overview of Martian Meteorites

The SNC (shergottite, nakhlite, chassignite) meteorite clan has been confirmed as originating from Mars on the basis of trapped noble gas compositions that match that of the Martian atmosphere [5]. These meteorites also plot along their own mass-dependent fractionation line in triple-oxygen isotope space, with a $\Delta^{17}O$ of ~0.3 [6]. There are, to date, 150 meteorite pairing groups of Martian origin. While the traditional classification system divides these meteorites into shergottites (basalts constituting 89% of Martian meteorites [7]), nakhlites (clinopyroxene-rich cumulates), and chassignites (dunites) [8], additional samples have contributed to the variety of the Martian igneous record. Allan Hills (ALH) 84001 is an orthopyroxene cumulate [9], Northwest Africa (NWA) 7034 and its pairs are polymict breccias [10], and NWA 8159 and NWA 7635 are augite-rich shergottites with unique ages [11,12]. Together, the crystallization ages of these meteorites roughly span the age of the planet (Figure 1).

The shergottites, which represent the overwhelming majority of Martian meteorites, are subdivided into additional categories based on petrological and geochemical characteristics. Basaltic shergottites contain pyroxene and plagioclase in varying proportions and span grain sizes from gabbroic to fine-grained. Olivine-phyric shergottites are similar to basaltic shergottites but contain olivine phenocrysts, megacrysts, or xenocrysts, and poikilitic shergottites are mafic to ultramafic plutonic rocks with large-grained olivine and pyroxenes [13]. Geochemically, shergottites are divided into enriched, intermediate, and depleted categories based on the level of depletion of their light rare earth elements (LREE) and their Sm-Nd and Hf-W radiogenic isotope systematics [14–16]. While the majority of shergottites are young (<600 Ma), NWA 7635 and 8159 are 2.4 Ga exceptions [11,12] which extend the age of shergottite volcanism to encompass nearly the entire Amazonian Period (3.0–0 Ga). The nakhlites and chassignites, on the other hand, are geochemically distinct from the shergottites and could represent melts and cumulates from a single differentiated igneous body [17,18] or from several lava flows or shallow sills associated with a single volcano [19,20]. Allen Hills 84001 is an orthopyroxene cumulate with an age of 4.09 Ga [9] that gives unique insight into the Noachian Period (4.1–3.7 Ga) of Martian history [9], during which liquid water precipitated secondary carbonate [21]. Finally, igneous clasts and zircons in the Martian regolith breccias NWA 7034 and its pairs record evidence of an ancient crust that must have formed on Mars within the first 20 Ma of the planet's formation [22].

Analyses from orbital and surface missions have greatly expanded the compositional diversity of Martian igneous rocks. With the exception of NWA 7034 and its pairs, Martian meteorites are all basaltic to ultramafic. Orbital measurements of volcanic provinces by Gamma Ray Spectroscopy (GRS) aboard Mars Odyssey show that the average Martian crust is basaltic to trachybasaltic [23,24]. The igneous rocks at Gusev Crater examined by the Mars Exploration Rover (MER) Spirit are roughly basaltic but have much higher ranges of total alkalis than the SNC meteorites [25]. The Mars Science Laboratory (MSL) has analyzed igneous rocks in Gale Crater and identified diorites, trachytes, trachyandesites, and quartz diorites [26]. Such diverse measurements suggest that Martian igneous geology is much more varied than is implied by the SNC meteorites, involving large degrees of differentiation and fractional crystallization.

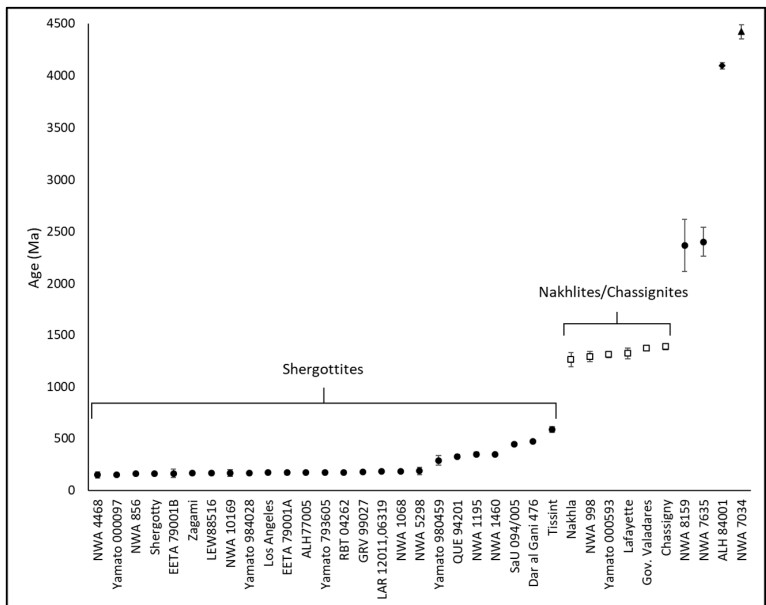

**Figure 1.** Crystallization ages of shergottites (black circles), nakhlites and chassignites (white squares), Northwest Africa (NWA) 7034 (black), Allan Hills (ALH) 84001 (black diamond), and NWA 7034 (black triangle). The age shown for NWA 7034 corresponds to the oldest measured igneous clasts. Data are from [11,12,19,22,27,28] and references therein.

## 3. Crystallization and Ejection Ages

The radiometric age dating of Martian meteorites has been used to constrain the timing of Martian core formation and early silicate differentiation into distinct magmatic reservoirs. The subsequent melting and possible interaction of these reservoirs created the petrogenetic diversity of the Martian meteorite suite [13]. Since Mars is a planet with a significant gravity well, large impacts and their associated shock pressures are required to loft material from the planet [29,30]. Most Martian meteorites are moderately to heavily shocked [31], resulting in partial melting and transformation of their plagioclase to the shock phase maskelynite. As the $^{40}Ar/^{39}Ar$ chronometer is sensitive to such disruptions [32], the $^{40}Ar/^{39}Ar$ dating of Martian material must be carefully evaluated in regard to crystallization ages. Likewise, because most Martian meteorites are desert finds, the $^{87}Rb/^{87}Sr$ chronometer is often disrupted by calcite deposition which contains terrestrial alkali and alkaline elements. Similarly, chronometers that employ the Sm-Nd system are susceptible to fluid mobilization and addition of these elements (as well as the rest of the rare earth elements) [4,33]. Figure 2 summarizes the $^{147}Sm/^{143}Nd$ dating that has been performed on Martian meteorites.

$^{147}Sm/^{143}Nd$, $^{87}Rb/^{87}Sr$, and $^{176}Lu/^{176}Hf$ dating of the shergottites has shown crystallization ages that cluster according to their geochemical enrichment. The enriched shergottites have ages between ~165 and ~200 Ma [34–42]. The intermediate shergottites have overlapping and older crystallization ages between ~150 and ~350 Ma [43–49]. Depleted shergottites are the oldest group, with crystallization ages between ~327 and ~2400 Ma [11,12,50–55]. Prior to two recent studies, the depleted shergottites were all thought to have been younger than ~600 Ma. NWA 7635, with a $^{147}Sm/^{143}Nd$ crystallization age of 2403 ± 140 Ma [12], and NWA 8159, with a $^{147}Sm/^{143}Nd$ crystallization age of 2300 ± 250 Ma [11], have greatly extended the potential magmatic history of the shergottites. Ejection ages for shergottites are all under 5 Ma [56] with the exceptions of Dhofar 019 (18 Ma). Depleted shergottites and NWA 7635 have been suggested to originate from the same igneous body based on their nearly identical ejection ages of ~1.1 Ma [12,56,57], and this suggests the existence of continuous Martian volcanism for half of the planet's history.

An additional complication in the Martian crystallization story is the "Old Shergottite Paradox." While the $^{147}Sm/^{143}Nd$, $^{87}Rb/^{87}Sr$, and $^{176}Lu/^{176}Hf$ chronometers show Amazonian ages for the shergottites, the Pb-Pb system has consistently yielded concordant ages older than 4 Ga [58–60]. Given that aside from some young basaltic lava flows, the Martian surface appears to be predominantly ancient (>4 Ga) [61], the Pb-Pb ages of shergottites have been suggested to reflect primary crystallization of these ancient terrains. This would argue that the younger ages produced by the other chronometers were a result of later disruptions such as impact resetting, fluid percolation, the drying of lakes, or even the combination of impacts and wet soil, creating superheated steam which would more effectively reset phosphate-based chronometers [60].

Recent studies have called the old ages of the shergottites into question, disputing the presence of any >4 Ga Pb-Pb "isochron" with crystallization age significance. Secondary ion mass spectrometry (SIMS) Pb-Pb measurements of maskelynite grains in ALH 84001 and some enriched shergottites, interpreted to reflect initial Pb isotopic compositions since plagioclase incorporates very little U, showed significantly differing values, presumably due to crystallization ages separated by ~4 Ga [62]. In-situ SIMS Pb-Pb analyses of different phases in Chassigny revealed a three-component mixing array between initial Pb, radiogenic Pb along a 1.39 Ga isochron, and an unsupported radiogenic reservoir [63]. This reservoir was heterogeneously present in phases whose major element compositions were otherwise homogenous, and its Pb composition plotted off the 1.39 Ga reservoir. Thus, it could not be explained by in-situ accumulation of radiogenic Pb and must represent non-igneous introduction of a likely crustal Martian reservoir. A hypothetical composition for this high $\mu$ reservoir is provided by the Martian regolith breccias [64], which the phases in Chassigny skew towards [63].

The $^{147}Sm/^{143}Nd$ and $^{87}Rb/^{87}Sr$ dating of nakhlites and chassignites have yielded crystallization ages around ~1300 Ma [27,65–68], with a mean age of 1340 ± 40 Ma [19]. The ejection ages of most nakhlites and Chassigny cluster around ~10 Ma [20,56]. Additionally, nakhlites and chassignites

share similar trace elements [19,69] and volatile-bearing phosphate [17] compositions. These lines of evidence and their similar petrologic texture suggest that they were formed in separate flows, sills, and dikes as part of the same overall igneous complex [19] whose magmatic source reservoir is distinct from that of the shergottites.

The discovery of additional unique Martian meteorites has further complicated the magmatic history derived from study of the SNC suite. The $^{147}$Sm/$^{143}$Nd ages determined for the orthopyroxene cumulate ALH 84001 have converged at ~4400 Ma [44]. However, $^{176}$Lu/$^{176}$Hf dating found a younger age of 4091 ± 30 Ma which concords with its $^{40}$Ar/$^{39}$Ar and U-Pb ages [9]. The older ages were likely due to extensive alteration, but controversy over the true crystallization age still exists.

NWA 7034 and its pairs, the Martian regolith breccias, represent the oldest age dates yet found in Martian meteorites. Secondary ion mass spectrometry (SIMS) U-Pb dating of zircons in monzonitic clasts within one of its pairs, NWA 7533, revealed a discordia line with two intercepts at 4428 ± 25 Ma and 1712 ± 85 Ma, suggesting that the igneous clasts in the meteorite are sourced from an evolved Martian crust from the first ~100 Ma of solar system history [70]. The 1.7 Ga age is similar to a 2.1 Ga $^{87}$Rb/$^{87}$Sr age [10], suggesting a major disturbance around that time. U-Pb dating of phosphates within the matrix also found a younger age of 1357 ± 81 Ga, while Pb-Pb analyses of feldspars in igneous clasts identified a high $\mu$ ($^{238}$U/$^{204}$Pb) reservoir at least 4428 Ma in age [64], providing further evidence for an ancient enriched crust. $^{147}$Sm/$^{143}$Nd dating of the igneous components of the regolith breccia provided additional confirmation with an age of 4420 ± 70 Ma [71]. Additional analyses of zircons within igneous clasts and matrix found two discrete sets of concordant U-Pb ages at 4431 ± 27 Ma and 1502 ± 98 Ma [72]. Cl-apatite from igneous clasts and matrix analyses in the same study revealed a U-Pb age of 1495 ± 88 Ma, suggesting a breccia-wide thermal event at that time such as an impact or volcanism.

The most precise zircon analyses conducted on the Martian regolith breccia, using acid dissolution and thermal ionization mass spectrometry (TIMS) instead of in-situ methods, yielded crystallization ages between 4476.3 ± 0.9 and 4429.7 ± 1.0 Ma [22]. Since the meteorite is only mildly shocked (<15 GPa) [73], $^{40}$Ar/$^{39}$Ar analyses could also be used to extract reliable age information other than shock ages, as with the rest of the SNC suite. $^{40}$Ar/$^{39}$Ar analysis of whole-rock fragments found several coherent plateau ages between 1319 ± 16 and 1191 ± 32 Ma [28]. Rather than thermal metamorphism resulting from an impact, which would cool in a few 10s of Ma or less [74], these plateaus, along with the overlapping U-Pb ages, suggest a protracted period of metamorphism that varied spatially and temporally, such as that resulting from contact metamorphism due to volcanism [28]. An even younger event of ~225 Ma or earlier was identified via U-Th-Sm/He chronometry, suggested to be an impact that resulted in the brecciation and consolidation of the meteorites in their current form [28]. Finally, an ejection event of ~5 Ma [75] brought the breccia into an Earth-crossing orbit. In summary, NWA 7034 and its pairs sample an ancient, evolved Noachian crust, distinct Amazonian volcanic and impact events, and a unique and recent ejection event, none of which are evident via the rest of the Martian meteorite suite.

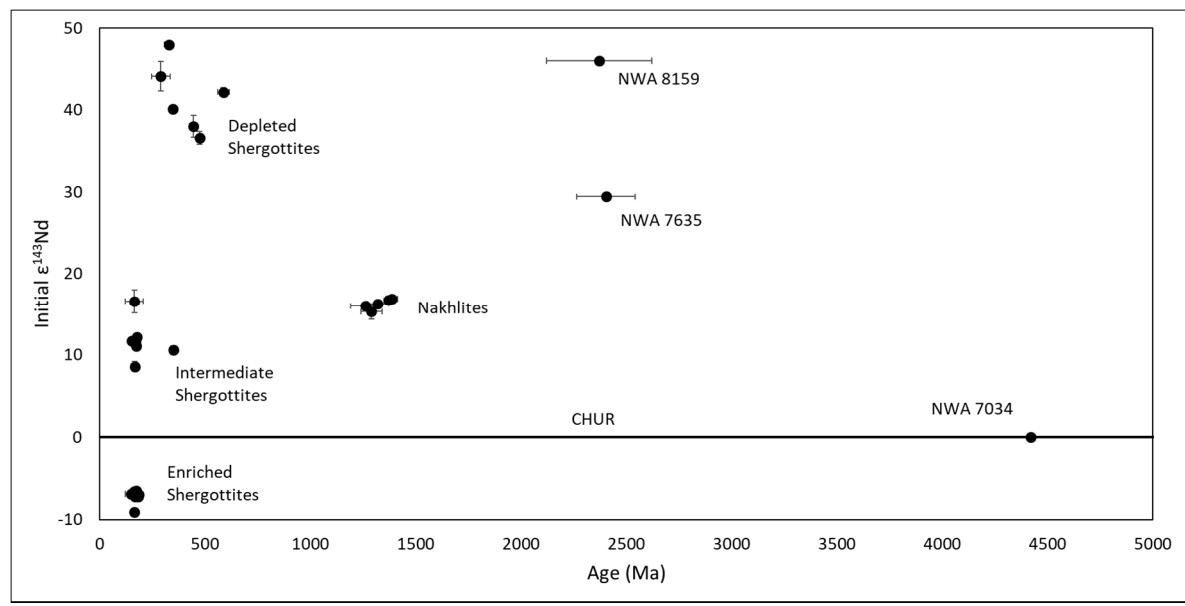

**Figure 2.** $^{147}$Sm/$^{143}$Nd ages and calculated initial εNd values for Martian meteorites [11,12,35–40,43–55,65–68,71,76,77].

## 4. Differentiation

Since they represent the only samples available from Mars, the Martian meteorite suite has been used in attempts to understand the large-scale magmatic processes that have shaped the planet's geologic history. For instance, the major and trace element compositions and isotopic systematics of shergottites have been used as constraints to test petrogenetic models of the magmatic differentiation of silicate Mars. One way to explain the geochemical diversity of shergottites is by assuming they formed by crustal assimilation and fractional crystallization (AFC) of mantle-derived magmas [78]. However, the shergottites' bulk $^{87}$Sr/$^{86}$Sr and $^{87}$Rb/$^{86}$Sr values have been shown to plot along the ~4.5 Ga isochron known as the basaltic achondrite best initial (BABI) line [39,79]. The AFC model would thus require the crustal assimilated material to have an age of ~4.5 Ga and remain undisturbed through the crystallization ages of all of the shergottites, which is unlikely [43] given the history of impact and volcanism experienced by the planet. In addition, the shergottites' incompatible trace element abundances, ratios, and isotopic compositions do not correlate well with their mineralogical or geochemical indices of differentiation, such as $SiO_2$ content or Mg# (molar Mg/(Mg + Fe)) [14]. The Martian regolith breccia, as its zircons and igneous fragments represent an enriched ancient crust, was likewise found to deviate from the shergottite mixing line in $\varepsilon^{143}$Nd-$\mu^{142}$Nd space (discussed further below), invalidating its role as an enriched end-member for shergottite crustal assimilation [80].

Another model invoked to explain Martian basaltic volcanism involves equilibrium and fractional crystallization of the Martian magma ocean (MMO) into cumulate piles and enriched residual liquids [14]. The cumulate piles were then melted and mixed in varying proportions with the residual liquids to produce the diverse parent liquids of the shergottites. This two-stage model proposed a mixing relationship between geochemically enriched and depleted shergottite source regions early in Martian geologic history, which would satisfy the constraints imposed by the shergottites' bulk Rb-Sr systematics [14]. The compositions of cumulate packages were constrained using the bulk composition of Mars [81,82] and petrological melting experiments at estimated pressures of the crystallizing MMO that included majoritic garnet as a near-liquidus phase [83]. Partial melting of the cumulate piles was able to reproduce the major element concentrations and radiogenic parent/daughter ratios of the calculated parent liquid of ALH 77005 [84]. Since this liquid was the most mafic of the Martian basalts, it was used as a least-differentiated end member in the model. More evolved compositions could then be reached through olivine fractionation. The trace incompatible element abundances of Martian

basaltic liquids were reproduced by including a small fraction of residual liquid, solidified in the final stages of MMO crystallization, in the partial melting phase.

The combinations of several radiogenic isotope systems have been used to constrain the results of differentiation modeling. The short-lived isotope $^{146}$Sm decays to $^{142}$Nd with a half-life of ~103 Ma and thus imparts a $^{142}$Nd anomaly on material that differentiated early enough in solar system history that it was still a live nuclide. (This age has recently been questioned by a study that found half-life of 68 Ma [85]. Despite this, the present paper assumes the canonical value in reporting literature ages. The IUGS-IUPAC recommends using both half-lives to calculate ages until the discrepancy is resolved [86]. For discussion see [87].) When $^{146}$Sm-$^{142}$Nd is combined with longer-lived isotopic systems such as $^{87}$Rb/$^{87}$Sr and $^{147}$Sm/$^{143}$Nd, the timing of both source differentiation and liquid crystallization can be constrained in a single model [14,52]. For example, the $\varepsilon^{142}$Nd (($^{142}$Nd/$^{144}$Nd$_{sample}$/$^{142}$Nd/$^{144}$Nd$_{standard}$ − 1) × 10,000) of the basaltic shergottite Queen Alexandra Range (QUE) 94201 was found to be high enough to require fractionation of its source from bulk Mars no later than 33 Ma after the planet's formation [52]. However, its calculated initial $\varepsilon^{143}$Nd (($^{143}$Nd/$^{144}$Nd$_{sample\ initial}$/$^{143}$Nd/$^{144}$Nd$_{CHUR\ at\ T}$ − 1) × 10,000) required that its source remained relatively inactive until its crystallization age of ~327 Ma because any intermittent melting would have made the $\varepsilon^{143}$Nd value too high [52].

Early models of shergottite source crystallization combined the $^{146}$Sm-$^{142}$Nd and $^{147}$Sm-$^{143}$Nd chronometers for multiple Martian samples by plotting their initial $\varepsilon^{143}$Nd values, recalculated at 175 Ma, the average crystallization age of several shergottites, against their present-day measured $\varepsilon^{142}$Nd values [50]. Several enriched and depleted shergottites formed an isochron that intersected with the chondritic uniform reservoir (CHUR) and converged on an age of ~4510 Ma, interpreted as the age of MMO differentiation from CHUR into geochemically enriched and depleted reservoirs. Other samples, such as the nakhlites and chassignites, plotted off this isochron, implying derivation from separate reservoirs or more complex igneous history [14].

The coupled $^{142}$Nd-$^{143}$Nd chronometer was later refined in regard to shergottites with higher precision measurements of $^{142}$Nd and additional sample measurements. An updated plot of enriched, intermediate, and depleted shergottites on the $\varepsilon^{143}$Nd vs. $\varepsilon^{142}$Nd graph formed a mixing line that missed the origin and thus is not likely to be an isochron (for an example of such a plot see Figure 3) [15]. Instead, the coupled isotopic systematics for depleted and enriched shergottites were interpreted to constrain earlier and later Sm/Nd fractionation events, at ~4535 Ma and ~4457 Ma, respectively. The calculated $^{147}$Sm/$^{144}$Nd of the shergottite sources was found to be higher for each meteorite than the $^{147}$Sm/$^{144}$Nd of the meteorites themselves, and this is the opposite of what is expected during partial melting since Nd is a more incompatible element than Sm [15]. Therefore, the source of the shergottites likely underwent a partial melting event immediately before melting to produce the shergottites, and the enriched and intermediate shergottites incorporated varying amounts of this depleted source and a distinct enriched source.

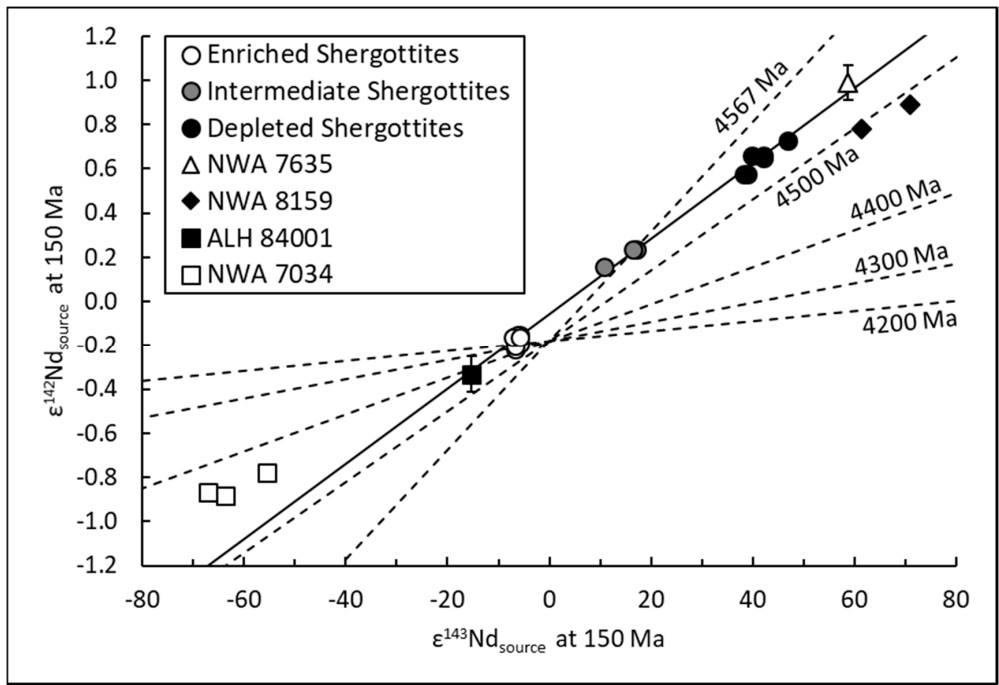

**Figure 3.** Calculated source $\varepsilon^{142}$Nd vs. $\varepsilon^{143}$Nd at 150 Ma for Martian meteorites. $\varepsilon^{142}$Nd values for ALH 84001 and the Martian regolith breccia sources are corrected for radiogenic growth of $^{142}$Nd. Solid black line is the shergottite source regression line (SSRL) whose slope reflects the age of shergottite source differentiation (4506 ± 6 Ma [87]). Dotted lines are model age isochrons which pass through the chondritic $\varepsilon^{142}$Nd value of −0.18 [15]. Since the SSRL does not intersect the model age origin, either it is not an isochron, or Mars' initial $^{142}$Nd composition was not chondritic [87]. The source of Martian regolith breccia does not intersect the SSRL, suggesting that the shergottite differentiation event was not a planet-wide event such as crystallization and overturn of a magma ocean. Data and models are from refs. [80,87] and references within. Errors are plotted when symbol is smaller than error.

Additional analyses of shergottites have cast doubts on the need for multiple melting and differentiation events. The linear mixing line [15] found between enriched and depleted shergottites in $^{147}$Sm-$^{144}$Nd space has been obscured by newer data [54,88] showing variation in bulk rock $^{147}$Sm/$^{144}$Nd in meteorites with similar calculated source $^{147}$Sm/$^{144}$Nd values [87]. Newer models showed that two separate three-stage petrogenetic models using shergottite bulk rock $^{142}$Nd, $^{147}$Sm, and $^{143}$Nd measurements converged on a source differentiation age of 4504 ± 6 Ma [87]. The first model formed an isochron by plotting the measured bulk rock $^{142}$Nd/$^{144}$Nd of the shergottites against their source $^{147}$Sm/$^{144}$Nd, calculated from measured $^{143}$Nd/$^{144}$Nd values. The second plotted the $^{142}$Nd/$^{144}$Nd of each meteorite against their present-day source $\varepsilon^{143}$Nd, calculated from each meteorite's initial $^{143}$Nd/$^{144}$Nd (see Figure 3). Importantly, neither model relied on an assumed initial $^{142}$Nd/$^{144}$Nd for bulk Mars, and both allowed for binary mixing between depleted and enriched shergottite end members early in Martian history.

The Sm-Nd data from nakhlites and chassignites have been shown to diverge significantly from any model age isochrons formed by the shergottites [89]. Thus, the variation in Sm and Nd isotopic compositions between shergottites and nakhlites cannot be attributed to a single differentiation event early in Martian geologic history. Likewise, the crustal breccia NWA 7034 diverges from the shergottite mixing line in $^{142}$Nd-$^{143}$Nd space, using either measured values from the meteorite or ones calculated for its source (Figure 3) [80]. This suggests that it does not represent an enriched crustal end member for shergottite differentiation, and it casts doubts on the idea that the shergottite mixing line is in fact an isochron.

The extinct $^{182}$Hf-$^{182}$W chronometer, which has a half-life of 9 Ma, has been coupled with the $^{142}$Nd/$^{144}$Nd system to refine the time scale of magmatic source differentiation [16]. The combined $^{182}$W-$^{143}$Nd chronometer showed that instead of forming a binary contemporaneous mixing line between depleted and enriched shergottite sources, the shergottite source differentiation was instead spread between ~20–25 and ~40 m.y. after solar system formation (Figure 4) [16]. This was due primarily to a larger range in $\varepsilon^{182}$W identified among depleted shergottites (+0.8 to +1.8). $^{142}$Nd measurements of NWA 7034 and ALH 84001 also showed that these samples reflected the most enriched and early to differentiate (~20–25 Ma) source reservoir yet identified on Mars [16]. The early differentiation of an enriched source coincides with modeling results obtained from U-Pb and Lu-Hf analyses performed on zircons in NWA 7034, which required an andesitic, rather than basaltic, crustal reservoir to differentiate from CHUR in order to satisfy the $^{176}$Lu/$^{177}$Hf values of concordant zircons [22]. The U-Pb ages of these zircons converged at 4475 Ma, while their Lu-Hf systematics suggested that an andesitic crust must have formed no later than 4547 Ma. This early crust was completely reworked by impacts, including the one likely to be responsible for the formation of the Martian crustal dichotomy [90].

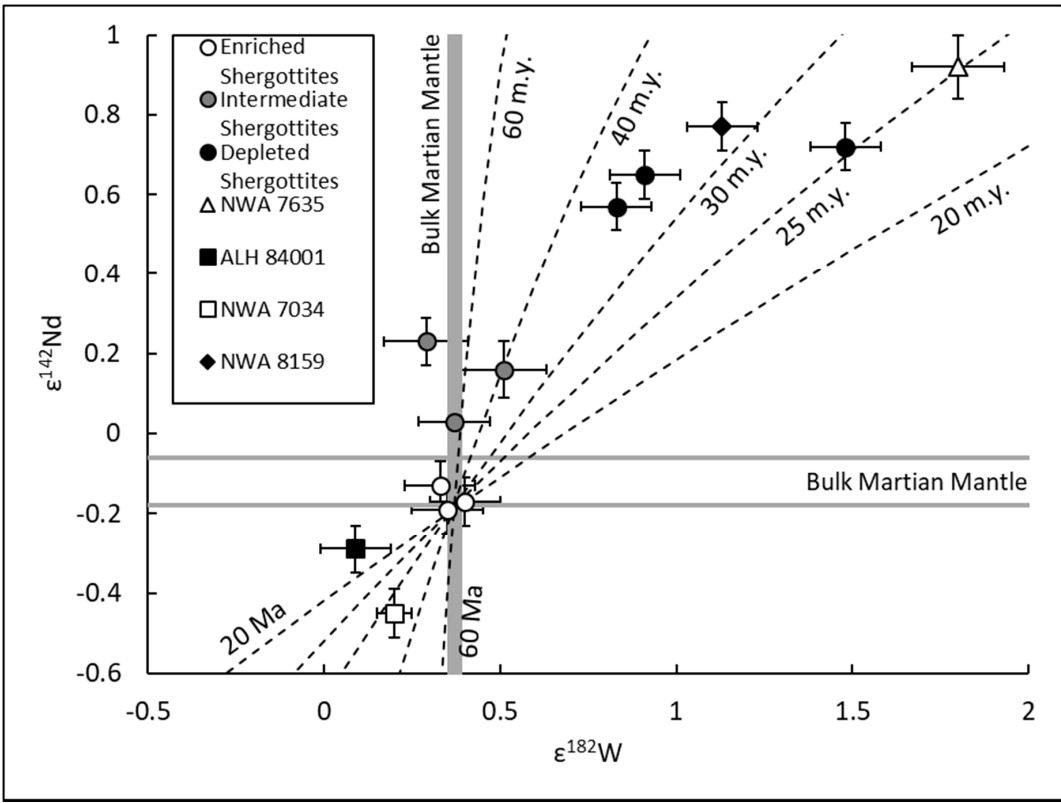

**Figure 4.** $\varepsilon^{182}$W vs. $\varepsilon^{142}$Nd for Martian meteorites. Grey horizontal lines are the ranges of potential Martian bulk $\varepsilon^{142}$Nd values [15,87], and grey vertical line is the range of Martian bulk $\varepsilon^{182}$W, deduced from these values and Martian meteorite $\varepsilon^{182}$W. Dashed lines represent model times for source differentiation after solar system formation. Data and model calculations from ref. [16] and references within.

As this early crust predates meteorite magmatic source differentiation, its age also suggests that this differentiation did not occur under the regime of a crystallizing magma ocean. Instead, the crystallization of the MMO must have happened earlier, resulting in a stratified mantle susceptible to overturn, decompression melting of cumulate material, and extraction of an evolved andesitic crust [22,91,92]. Alternatively, the MMO may not have existed at all, as a recent study has found hydrogen isotopic heterogeneities within Martian mantle source reservoirs [93]. A series of later differentiation events must have formed the source reservoirs of the other Martian meteorites. The nakhlites, which have

significantly more radiogenic $\varepsilon^{182}$W values (~3) [94,95], must have originated from a separate but roughly coeval magmatic source. The positive $\varepsilon^{182}$W and $\varepsilon^{142}$Nd values of SNC meteorites require that their sources formed from a garnet- and clinopyroxene-bearing mantle, since these phases fractionate Hf/W as well as Sm/Nd [96]. The ongoing magmatism resulting from these differentiation events, along with impacts, completely resurfaced the planet.

## 5. Core Formation

Because W is a much more siderophile element than Hf, it fractionates completely from Hf during metal–silicate separation. The timing of the core formation of planetary bodies is constrained by measuring $^{182}$W anomalies from primitive chondritic values in silicates derived from material that underwent core formation while $^{182}$Hf was still extant. Measuring the $\varepsilon^{182}$W of Martian meteorites showed a small range in values for the shergottites (0.3 to 0.7) and a distinct and uniform value of ~3 for the nakhlites [94]. By plotting $\varepsilon^{142}$Nd values against $\varepsilon^{182}$W values for the shergottites and solving the best-fit line for chondritic $^{142}$Nd, a primitive Martian mantle (PMM) $\varepsilon^{182}$W of 0.34 ± 0.7 was determined (see Figure 4). This value allowed for a two-stage model age for the Martian core formation of 11.6 ± 0.4 m.y. after solar system formation [94]. Other estimates for Martian core formation using W isotopes have ranged from ~3 to ~15 m.y. after solar system formation [95,97].

A more recent study has found more variation in $\varepsilon^{182}$W among the shergottites and that this variation correlates with $\varepsilon^{142}$Nd [16]. However, the new measurements did little to modify the bulk Mars $\varepsilon^{182}$W value, since the Martian meteorites whose $\varepsilon^{142}$Nd spanned the range of estimated bulk Mars $\varepsilon^{142}$Nd values [15,87] were relatively constrained in $\varepsilon^{182}$W. Thus, the new PMM $\varepsilon^{182}$W was found to be +0.37 ± 0.04 (Figure 4), within the error of the old one. Combining this value with the bulk Martian $^{180}$Hf/$^{184}$W ratio of 4.0 ± 0.5 [98] yielded a two-stage model age for core formation of 4.1 ± 2.7 m.y. after solar system formation (Figure 4) [16]. This is consistent with the accretion timescale of a stranded planetary embryo, in which Mars attained half of its mass in ~2 m.y. or less [16,98].

## 6. Conclusions

It is clear at this point that the 150 unpaired Martian samples that are currently available for study vastly undersample the igneous history of the planet Mars. The identification of petrologically diverse and evolved material by rovers suggests that the shergottites, nakhlites, chassignites, ALH 84001, and the Martian regolith breccias only offer a small window into the magmatic evolution of the planet (Table 1), and thus the need for sample return missions is highlighted by the study of Martian meteorites. There is now significant petrological and geochemical evidence that both the nakhlites and chassignites as a group, as well as the depleted shergottites, both originate from their own unique magma bodies. The Martian regolith breccias also sample a unique magmatic reservoir and possibly an ancient enriched crust that was later modified by impacts. These sources must have been separated since the earliest differentiation of Mars shortly after the birth of the solar system and stayed separate through the crystallization of the Martian igneous suite.

**Table 1.** Summary table of Martian meteorite ages, rock types, and geochemistry.

| Meteorites | Ages | Rock Type(s) | Geochemistry |
|---|---|---|---|
| Shergottites | ~150–600 Ma; ~2400 Ma | Gabbro, diabase, basalt | Depleted to Enriched in LREE |
| Nakhlites | ~1300 Ma | Augite and olivine cumulate | Enriched in REE and LREE |
| Chassignites | ~1300 Ma | Dunite | Enriched in LREE |
| NWA 8159 | ~2300 Ma | Augite basalt | Depleted in LREE |
| ALH 84001 | ~4100 Ma | Orthopyroxenite | Nearly chondritic REE |
| NWA 7034 and pairs | ~4500 Ma | Polymict igneous breccia | Enriched in REE and LREE |

The need to better define the magmatic reservoirs present on Mars is another urgent reason to increase sample diversity. The Martian mantle is clearly heterogeneous, and its igneous products show a great deal of diversity in isotopic systematics, such that the timing for major events in the planet's history is likely to be subject to change (Figure 5). This diversity is reflective of a silicate Mars that was inefficiently mixed during its early history. More unique magmatic source reservoirs are likely waiting to be discovered with additional samples from the planet. The wealth of research that the discovery of the Martian regolith breccias has generated is a testament to this, and new potential Noachian and Hesperian samples will likely again completely redefine our understanding of Martian igneous history.

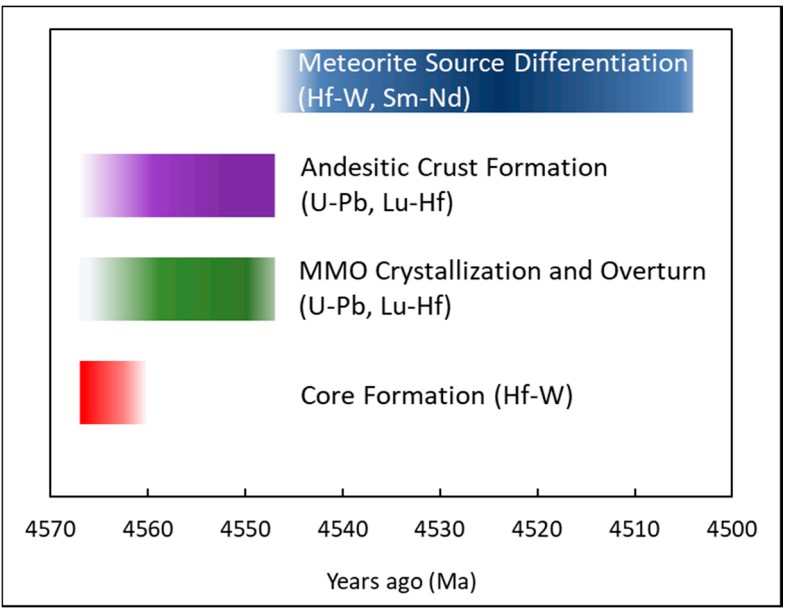

**Figure 5.** Summary of early Martian events based on various chronometers [16,22,87,89,94,99,100].

**Author Contributions:** Conceptualization, methodology, software, validation, formal analysis, investigation, resources, data curation, writing—original draft preparation, visualization, supervision, project administration, funding acquisition, Z.V.; writing—review and editing, Z.V., C.A. All authors have read and agreed to the published version of the manuscript.

**Funding:** This research received no external funding.

**Acknowledgments:** The authors are thankful for two anonymous reviewers for helping to improve the quality of this manuscript.

**Conflicts of Interest:** The authors declare no conflict of interest.

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
