# Peer review of "Constraints on Martian Chronology from Meteorites"

_geosciences, doi:10.3390/geosciences10110455_

Round 1

Reviewer 1 Report

See attached file

Author Response

Please see attached file of replies.

Reviewer 2 Report

Review of Constraints on Martian Chronology from Meteorites by Zoltan Vaci and Carl Agee

I have read this manuscript as well as the previous review of an older version of this manuscript.  Like the previous reviewer—and even after some improvements—I also have concerns about the organization of the manuscript.  Why are crystallization and ejection ages discussed together when they are so fundamentally different?  Why is differentiation a topic to be considered here except in the context of the ages of those differentiation events?  Why discuss core formation last when it happens early in the history of Mars?

The importance of uncertainties (analytical and model-derived) is not sufficiently discussed in the text.  A plot like Figure 1 would be so much more useful if it had error bars, and if it showed the ranges of measured dates that go into each point.  NWA 7034 breccia ages cluster in two (concordia or discordia intercept) groups, but the dates do vary within those groups and that uncertainty is critical for comparing the younger subset of NWA 7034 U-Pb ages with Ar-Ar ages of the whole rocks.  The agreement or lack thereof between U-Pb and Ar-Ar of those samples results in a disputed requirement for an additional long-lived metamorphic event in the history of the breccia.

All discussions of model ages would greatly benefit from figures demonstrating how the data lead to the conclusions, and those plots are strangely missing throughout the manuscript (though they are repeatedly discussed in absentia).  This is especially important for examples where there is debate about the significance of different chronometers, or where there are multiple interpretations of the same data.  But in general, the presentation of model ages without the critical diagrams used to produce those ages does not serve the reader well.  I know the additional work required to replot the data from previous works is not trivial, but I think it is essential to add those figures if the authors want the readers to have a complete understanding of the calculated ages.

Figures 2 and 3 appear to not be referred to in the text at all.  Why present these figures instead of the figures described in the text that would increase the readers understanding of the complicated model ages that the author spend so much time explaining, and are so clearly important?  Figure 3 is not only not called out in the main text (at least so far as I can find), but it has an extremely uninformative caption.

Overall, I think this manuscript is in need of substantive revision before it would be acceptable for publication.  I would consider reviewing a revised version.

I have some additional minor comments, which I’ve listed below by line number.

  1. Delete “age”.The words “age” and “date” have different meanings and should not be used in combination.

  1. Does “diabasic” add anything to this sentence?

  1. This sentence could be rewritten to improve clarity, perhaps omitting the word trends since you are discussing process not plotting.

  1. In figure 1, use different symbols and colors to aid viewing by people who are color-blind, choosing colors that are easier to resolve from each other.The black and dark grey are too similar to be resolved by color description alone, so be sure to emphasize in the caption that the shapes are different as well.  I also think this plot would be more useful if it had error bars or if the data was depicted as a series of box plots.  Otherwise who is to say what is within error and what is not?

  1. Are all uncertainties two sigma?Please indicate and be consistent.

  1. I think that the existence of young and old volcanic rocks does not necessarily mean that volcanism was long-lived, even on a planetary scale.

  1. The nakhlite-chassignite group is represented here by a single age with ± 40 Ma uncertainties.Assuming that this is two sigma of the mean, what fraction of that range can be explained by analytical uncertainties, and how much is unexplained scatter?  Are we really justified in saying that the array of dates represents a single age?

  1. Maybe the discovery of new samples that don’t fit the simple models is an indication that we must stop trying to model Mars as if it were analogous to a single volcano on Earth.Perhaps Mars is complicated, and no single model can explain all the rocks, or even all of any given subgroup.  I am trying to imagine collecting a random assortment of igneous samples from the Earth and then trying to interpret them together.

  1. Was this really U-Pb dating, or was it Pb-Pb?I have not read that paper but feldspars are notoriously low in U, and are generally used to derive initial, unradiogenic Pb isotope compositions.  Those can then be used to construct model ages.  But direct U-Pb dating of feldspars is unknown to me.

  1. Do you mean chemical abrasion TIMS U-Pb geochronology?

  1. Is this a typo of U-Th-Sm/He?

  1. I would err on the side of under-interpreting the whole-rock, step-heating Ar-Ar data, as it is not clear (to me at least) that the proposed thermal history needs to be distinct from what is recorded by U-Pb.

  1. This section would benefit from a couple of figures showing the data with the two different models.

  1. Explain why it is unlikely to have an old, preserved reservoir, especially given the evidence presented on line 251 that old rocks are known to have been preserved.

  1. The authors need to do a better job tying these petrologic constraints to the isotope geochemistry and geochronology that this manuscript is focused on.

  1. There is recent evidence questioning the canonical existence of an MMO (Barnes et al. 2020).

  1. This is offered as an alternative to the model in the previous paragraph, but the authors do not explain how this can result in Rb-Sr systematics that plot along BABI.

  1. The official IUGS-IUPAC recommendation is to use both half-lives and report ages for both (Villa et al. 2020 GCA).

  1. Is this a similar conclusion to that of line 170?

  1. It is strange to be discussing plots in other papers, instead of demonstrating these important points with figures in this manuscript.

  1. Another discussion of a figure that is not in the manuscript.It is clearly important, so it should be replotted from the original data.

  1. The word “spread” can be interpreted in several ways.Is this due to scatter in the data that is explained by uncertainties in the measurements?  Is this due to precise data that do not fall on the model line? Is this two populations at ~20-25 and 40 m.y.?  Also, note that the use of Ma (or Ga) is reserved for ages relative to present day, and that other periods of time should use the abbreviation m.y (or b.y.) along with an explicit statement of what the date is relative to.

  1. I think I understand lines 254 and 255, but I do not see how one follows the other, and I think this would benefit from some clarification.

  1. I think you don’t mean 11 Ga after solar system formation, but the notation should be m.y. not Ma.The same is true for line 272.

  1. Here is another example where having a figure for the reader to refer to would be useful.

  1. m.y. not Ma

  1. I agree with this statement but it seems inconsistent with the attempts to explain the large suites of samples with single models.

  1. Not destroyed.Modified?

  1. It is not clear what the authors mean by a “set”, but it sounds like a straw man argument:Why would any planet be comparable in complexity to a (sub)set of rocks from that (or any other) planet?

  1. Or Mars never completely melted, as suggested by Barnes et al. 2020.

  1. Is this a reference to a specific set of samples?I think it is, and that the authors should be more explicit.

Author Response

Please see attached file with replies.

Round 2

Reviewer 2 Report

Review of Vaci and Agee “Constraints on Martian Chronology from Meteorites”

This is my second review of this manuscript, and it is much improved from the first version.  The authors did not agree with every comment I made, but they made a good faith effort to incorporate the changes I requested, including greatly improved figures.

This manuscript is ready for publication, except for the following small suggestions:

  1. “does not incorporate U” might be better phrased as “incorporates very little U”

Figures 3 and 4.  Please add a key to these figures, as figures are often used in talks without the caption and they will be more useful if they are self-contained.

Author Response

The authors thank the reviewer for their very helpful feedback.  The two minor suggestions have been implemented.

This manuscript is a resubmission of an earlier submission. The following is a list of the peer review reports and author responses from that submission.

Round 1

Reviewer 1 Report

See file

Reviewer 2 Report

I don`t think that this manuscript has enough fruitful thinking to be published as it is. I suggest and recommend, the author to expand the discussion, including more literature, reviewing better the existed (and recently published) geochemical models for the different martian meteorites. In that case, i don`t want to provide any particular comment for the manuscript, since, it has to be updated at most of its sections. 

The introduction should be revised a lot. for example, there is a fresh fall like Tissint, with a lot background information and published papers, which is not considered. 

In general, i suggest the author to take the time, and prepare a robust and longer review paper (at least expand it by 100%), and not the current short version. He has to put more critical thinking!